# Quality and Functional Characterization of Acetic Acid Bacteria Isolated from Farm-Produced Fruit Vinegars

Sun-Hee Kim , Woo-Soo Jeong , So-Young Kim and Soo-Hwan Yeo *

Fermented and Processed Food Science Division, Department of Agrofood Resources, NIAS, RDA, Wanju 55365, Republic of Korea
* Correspondence: yeobio@korea.kr; Tel.: +82-63-238-3609; Fax: +82-63-238-3843

**Abstract:** Acetic acid bacteria (AAB) form a bacterial film on the surface of alcoholic solutions and ferment ethanol to acetic acid while also producing bioactive compounds. To discover functional AAB for industrial use, we isolated and selected strains from farm-produced vinegars using a $CaCO_3$-containing medium. The seven isolated strains belonged to *Acetobacter cerevisiae* and *Acetobacter pasteurianus*. These strains were tolerant to ethanol concentrations up to 12% (*v/v*). Acidification was seen for GHA 7, GYA 23, JGB 21-17, and GHA 20 strains at a growth temperature of 40 °C. The seven AAB isolates had strong antibacterial activity against *Staphylococcus aureus*. Antioxidant activity, as assessed using the DPPH and ABTS assays, was two- and four-fold higher than that for the negative control (1% acetic acid), respectively. We also observed 91.3% inhibition of angiotensin-converting enzyme activity for the KSO 5 strain, which was higher than that for the positive control, 0.1% captopril (76.9%). All strains showed complete inhibition of $\alpha$-glucosidase, except JGB 21-17 and GHA 7, which showed 98.3% inhibition. Our work suggests the usefulness of the selected strains as seed strains for the highly efficient production of functional vinegar and illustrates the identification of useful functional characteristics on a scientific basis.

**Keywords:** acetic acid bacteria; alcohol tolerance; antioxidant activity; angiotensin-converting enzyme inhibition; $\alpha$-glucosidase inhibition



## 1. Introduction

Fermented foods with high nutritional value and scientifically demonstrated health benefits continue to attract attention. Multiple biochemical changes occur during fermentation that may increase nutritional value and lead to the production of bioactive metabolites. Vinegar has long been used to treat diseases in both the East and West [1]. The beneficial properties of vinegar include antioxidant [2], antitumor [3], hepatoprotective [4], antidiabetic [5], and antimicrobial [6] activities. Vinegar production by the action of acetic acid bacteria (AAB) involves the formation of a bacterial film on the surface of an alcoholic solution that ferments ethanol to acetic acid, in parallel with the generation of gluconic acid, glucuronic acid, polyphenols, vitamins, amino acids, and hydrolytic enzymes [7,8].

Acetic acid-producing bacteria are strictly aerobic Gram-negative bacteria of the *Acetobacteraceae* family that inhabit warm and humid sites of flowers, fruits, insect digestive tracts, and fermented foods, including vinegar, kefir, and kombucha [9–12]. They produce organic acids, aldehydes, and ketones through incomplete oxidative fermentation of sugars, alcohols, and sugar alcohols [13]. Acetic acid bacteria are industrially relevant microorganisms that have been used for the production of fermented foods, including vinegar, cosmetics, and medicines, and in the development of biofuel cells [12,14,15]. One strength of AAB is the ability to use less biomass while producing large amounts of acetic acid as compared to other bacteria that form organic acids [16]. The primary taste and aroma of vinegar products arise from AAB fermentation [17,18]. Among AAB, 19 genera have been reported to date, including *Acetobacter*, *Gluconobacter*, *Gluconacetibacter*, and



*Komagatacetibacter* [19]. *Acetobacter* is the main genus used in industrial vinegar production because of its high resistance to ethanol and acid [20,21].

Recently, as awareness of the benefits of vinegar has increased, more attention has been paid to research related to AAB in vinegar production. Investigations on the isolation and identification of AAB from alcoholic and acidic environments like vinegar, wine, cocoa bean, sugar cane, beer, fruit, and flower have been reported [22–27]. Moreover, with the implementation of the Nagoya Protocol in August 2018, the need for developing technology for resource recovery and formulation of indigenous bacteria has emerged to replace that of imported AAB. However, most studies have only demonstrated limited aspects of vinegar production by AAB, based on the characterization of products, with none focusing on the bacteriological properties and bioactivities of AAB. Therefore, to reduce royalty payments associated with the use of imported bacteria and contribute to the localization of AAB, we sought to discover functional AAB amenable for use in the processes for differentiation of vinegar and enhancement of its quality and to identify the bacteriological characteristics useful for industrial application. We isolated AAB from farm-produced fermented vinegars and characterized their bioactivities in terms of antibacterial, antioxidant, antihypertensive, and antidiabetic effects.

## 2. Materials and Methods

### 2.1. Collection and Preparation of Sample

Samples of 8 types of farm-produced vinegars were obtained from Gangwon-do, Gyeonggi-do, Jeollabuk-do, and Gyeongsangbuk-do provinces in the Republic of Korea. All samples were stored at 4 °C and screened for AAB isolates within 2–3 days of collection.

### 2.2. Isolation of Bacterial Strains

The AAB strains were isolated from farm-produced vinegars (Table 1) by plating them on YGC agar (5 g/L yeast extract, 30 g/L glucose, 10 g/L $CaCO_3$, 4% (*v/v*) ethanol, 2% (*w/v*) agar) [28]. Samples (100–200 µL) of the different vinegars were spread onto YGC agar plates and incubated at 30 °C for 3 days under aerobic conditions. Representative colonies with a clear halo, indicative of the dissolution of $CaCO_3$ in the medium by the acetic acid produced, were picked from the plates. These selected colonies were streaked onto fresh plates and used in further experiments. Purified strains were stored at −80 °C in YGC broth with 80% glycerol. As control strains, the following 3 strains obtained from the Korean Agricultural Culture Collection (KACC) center were used: *Gluconacetobacter saccharivorans* CV1, KACC 17057; *Acetobacter pomorum*, KACC 11998; and *Acetobacter syzygii*, KACC 12233.

**Table 1.** Information on the eight kinds of vinegar collected.

| Region | Origin (Vinegar) | Sample | PD [1] | CD [2] | Vessel |
|---|---|---|---|---|---|
| Gangwon-do | Apple | GHA | 11. 10. '20 | 03. 11. '21 | PET [3] |
| Jeollabuk-do | Korean blackberry (*Rubus coreanus*)_20 | JGB20 | 09. 25. '20 | 03. 20. '21 | Pottery |
| | Korean blackberry (*Rubus coreanus*)_21 | JGB21 | 01. 25. '21 | 03. 20. '21 | Pottery |
| | Apple | JGA | 04. 25. '20 | 03. 20. '21 | Pottery |
| Gyeonggi-do | Magnolia berry (*Schisandra chinesis*) | KSO | 01. 01. '21 | 03. 29. '21 | Glass |
| | Pineapple | KSP | 01. 10. '21 | 03. 29. '21 | Glass |
| | Tomato | KST | 12. 18. '20 | 03. 29. '21 | Glass |
| Gyeongsangbuk-do | Apple | GYA | 03. 16. '21 | 04. 26. '21 | Pottery |

[1] Production date (PD) of vinegar. [2] Collection date (CD) of vinegar. [3] Polyethylene terephthalate.

### 2.3. Spot Plate Assay

To accurately compare the alcoholic stress responses of the different strains, 2-fold serial dilutions of the cells were prepared in a liquid medium in 1.5 mL Eppendorf tubes. Bacteria in the exponential phase of growth were diluted to an OD 660 nm of 0.2. Two microliter aliquots of the bacterial suspension diluted to OD 600 nm of 0.2, 0.1, 0.05, and 0.025 were spotted on plates, which were incubated at 30 °C for 4 days.

### 2.4. Bacterial Culture Condition and Preparation for Analysis

The AAB strains were cultured in a liquid medium comprising 5 g/L yeast extract, 5 g/L glucose, 1% (*v/v*) glycerol, 0.2 g/L MgSO$_4$·7H$_2$O, 1% (*v/v*) acetic acid containing 5% (*v/v*) ethanol at 30 °C for 7 days at 150 rpm using a shaking incubator (MMS-210; EYELA, Tokyo, Japan) under aerobic conditions. For analysis, the culture broth was centrifuged at 10,000× *g* for 5 min, followed by the collection of the supernatant.

### 2.5. Identification of Bacterial Strains

Strain identification was performed via polymerase chain reaction (PCR) amplification using 16S rRNA gene sequencing. DNA was extracted using an AllPrep PowerFecal DNA/RNA Kit (Qiagen, Hilden, Germany), and the 16S rRNA gene region was amplified using the universal primers 27F (5′-AGAGTTTGATCCTGGCTCAG-3′) and 1492R (5′-GGTTACCTTGTTACGACTT-3′). PCR was performed on an ABI PRISM 3730xl (Applied Biosystems, Waltham, MA, USA), with reaction parameters of 30 cycles of denaturation at 96 °C for 10 s, annealing at 50 °C for 10 s and extension at 60 °C for 3 min. The sequences obtained were aligned using the Advanced Basic Local Alignment Search Tool of the National Center for Biotechnology Information. Phylogenetic analyses were performed using the MEGA software (version 6.0, https://www.megasoftware.net). A phylogenetic tree was constructed from alignments using the neighbor-joining method, and the reliability of the inferred trees was assessed using a bootstrap test [29].

### 2.6. Acetic Acid Production

Isolated AAB bacteria were evaluated for the ability to produce acetic acid on a YGC medium containing different concentrations of ethanol (3, 5, 7, 9, 10, 12, and 15% (*v/v*)) using the potency index (PI) for assessment [30]. After incubation at 30 °C for 96 h under aerobic conditions, a clear zone formed in the medium, indicating the production of acetic acid. The clear zone diameter (Supplementary Figure S1) was then used to assess the potency of each AAB. The diameters of the colonies formed by the isolates and the surrounding clear zones were measured, and the PI was determined according to the following formula:

$$\text{PI} = \frac{\text{Diameter of the clear zone (mm)}}{\text{Diameter of the bacterial colony (mm)}} \qquad (1)$$

Bacteria with the highest PI were selected. To analyze the ability of AAB to produce acetic acid, a liquid medium was used containing 5 g/L yeast extract, 5 g/L glucose, 1% (*v/v*) glycerol, 0.2 g/L MgSO$_4$·7H$_2$O, 1% (*v/v*) acetic acid containing different concentrations of ethanol (3, 5, 7, 9, 10, 12, and 15% (*v/v*)).

### 2.7. Bacterial Growth and Titratable Acidity

The growth of AAB was measured based on their optical density at 660 nm using a spectrophotometer (Gen5TM, BioTek, Winooski, VT, USA). The total acidity was determined by titration with 0.1 N NaOH using 1% phenolphthalein as an indicator. The volume of NaOH used in the titration was expressed as the titratable acidity (%) for neutralizing acetic acid, as presented in the following equation:

$$\text{Titratable acidity (\%)} = \frac{0.1\text{N NaOH (mL)} \times 0.006 \times 100}{\text{sample amount (mL)}} \qquad (2)$$

where 0.006 is the acetic acid equivalent.

### 2.8. Effect of Temperature on Growth

To analyze the effect of temperature on acetic acid production, a single colony was transferred to YGC agar containing 5% (*v/v*) ethanol. Plates were grown at 10, 20, 30, or 40 °C for 7 days under aerobic conditions, and the diameter of the clear zone was measured using a digital caliper at 3-day intervals.

### 2.9. Antibacterial Activity

The antibacterial activity of selected AAB strains against harmful Gram-positive (*Bacillus cereus*, KACC 10004; *Staphylococcus aureus*, ATCC 6538) and Gram-negative (*Escherichia coli*, KCTC 1309; *Salmonella typhimurium*, KCTC 41028) bacteria purchased from the Korean Collection for Type Cultures (KCTC, KACC) and American Type Culture Collection (ATCC) was investigated using an agar diffusion method on solid medium. The test strains were activated for 16 h in tryptic soy broth (17 g/L pancreatic casein digest, 3 g/L papain soybean digest, 2.5 g/L glucose, 5 g/L NaCl, 2.5 g/L $K_2HPO_4$), and 1:1000 dilutions of each test strain were then spread on a 0.6% tryptic soy agar plate. Sterile 8 mm disks (Whatman PLC, Maidstone, UK) were placed on the agar, and 40 μL of selected AAB strain cultures (optical density at 660 nm = 0.5) was inoculated onto the disks. After incubation for 24 h at 30 °C, the diameter of each zone with clear inhibition was determined. Three replicates were used. Positive control values were obtained using 12.5, 25, and 50 mg/mL of acetic acid.

### 2.10. Antioxidant Activity

The antioxidant activity of AAB strains was determined using 2,2-diphenyl-1- picrylhydrazyl (DPPH, Sigma-Aldrich, St Louis, MO, USA) and 2,2′-azino-bis-(3-ethylbenzothiazoline-6-sulfonic acid) (ABTS, Sigma-Aldrich) assays. The determination of the radical scavenging activity of the samples was carried out using a previously reported method [28,31] with slight modifications. For the DPPH assay, a stock solution of 0.4 mM DPPH in absolute ethanol was prepared, and a working DPPH solution was prepared by dilution with absolute ethanol to an absorbance of 0.95–0.99 at 525 nm. For each measurement, a 200 μL sample (standard dilution or ethanol blank) was mixed with 800 μL of working solution, and the solution was incubated in the dark for 90 min. The absorbance at 525 nm was then determined. For the ABTS assay, a stock solution of 2.6 mM $K_2S_2O_8$ and 7.4 mM ABTS diammonium salt was prepared in deionized water ($dH_2O$) and incubated in the dark for 24 h; thereafter, an ABTS working solution was prepared by dilution to an absorbance of 0.67–0.73 at 732 nm with phosphate-buffered saline, pH 7.4. The samples (50 μL standard dilution or $dH_2O$ blank) were mixed with 950 μL ABTS working solution for each measurement. The absorbance at 732 nm was then determined. All measurements were performed in triplicate using a microplate reader (SpectraMax M2, Molecular Devices, San Jose, CA, USA). The antioxidant activity in each case was determined relative to the control:

$$Antioxidant\ activity\ (\%) = 100 - \frac{A(sample)}{A(control)} \times 100 \tag{3}$$

### 2.11. Angiotensin-Converting Enzyme (ACE) Inhibition

Inhibition of angiotensin-converting enzyme (ACE) activity was determined using the method of Cushman and Cheung [32] with slight modifications. The method is based on the liberation of hippuric acid from hippuryl-L-histidyl-L-leucine (HHL) by ACE. For the assay, 50 μL of sample supernatant was mixed with 50 μL 2.5 mM HHL in 450 mM sodium borate at pH 8.3 and incubated at 37 °C for 5 min. The mixture was subsequently incubated at the same temperature for 40 min with 50 μL ACE (10 mU/mL). The reaction was terminated by the addition of 250 μL of 1 N HCl. Ethyl acetate (1.5 mL) was added, and the sample was mixed for 30 s. The sample was centrifuged at 10,000× *g* for 5 min, and the supernatant (1.0 mL) was dried in a heating block at 100 °C and subsequently dissolved

in 1.0 mL dH$_2$O. The absorbance at 228 nm was determined using a UV spectrophotometer (BioTek). An average of 3 readings was used to calculate ACE inhibition (%):

$$\text{ACE inhibition rate (\%)} = 1 - \frac{\text{S} - \text{SB}}{\text{C} - \text{CB}} \times 100 \tag{4}$$

where S is the sample absorbance in the presence of an ACE inhibitor, C is the control absorbance with dH$_2$O, and SB and CB are the absorbance readings for the sample and control blanks without ACE, respectively. Captopril (0.1%, (*v/v*)) was used as the positive control.

### 2.12. α-Glucosidase Inhibition

Inhibition of α-glucosidase activity was determined using an α-glucosidase assay kit (Cat. No. MAK123, Sigma-Aldrich), which is based on the release of *p*-nitrophenol by hydrolysis of *p*-nitrophenyl-α-D-glucopyranoside (α-NPG). A lack of detected activity was taken as 100% inhibition. For the assay, Master Reaction Mix (200 μL assay buffer, 8 μL α-NPG substrate) was mixed with 20 μL AAB supernatant and incubated at 37 °C for 20 min. The absorbance of *p*-nitrophenol was measured at 405 nm, and acetic acid (12.5 mg/mL) was used as a positive control. Inhibition of α-glucosidase activity was calculated as follows:

$$\alpha - \text{Glucosidase inhibition rate (\%)} = 1 - \frac{\text{A405(final)} - \text{A405(initial)}}{\text{A405(calibrator)} - \text{A405(water)}} \times 100 \tag{5}$$

where A405 (calibrator) is the absorbance of the calibrator at 20 min, and A405 (water) is the absorbance of water at 20 min.

### 2.13. Statistical Analysis

Three replicates of each experiment were carried out, and the data are reported as means ± standard deviations (SDs). Statistical analysis was performed through a 1-way analysis of variance using the Statistical Analysis System, v7.1 software (SAS Institute, Inc., Cary, NC, USA) and Duncan's multiple range test. A value of $p < 0.05$ was considered significant.

## 3. Results and Discussion

### 3.1. Identification of Collected Vinegars

Among the eight types of fruit vinegar collected, apple vinegar from Hongcheon, Gangwon-do, was made from a wine with 9.5% (*v/v*) ethanol produced from 12 Brix apple juice samples. This vinegar was fermented at room temperature (25–28 °C) in a PET container and had 4.5% acidity. Magnolia berry (*Schisandra chinesis*) vinegar from Seongnam, Gyeonggi-do, was prepared from 3-year-old magnolia berry syrup and had 4.42% acidity. Pineapple and tomato vinegars from Seongnam were prepared from fruit wine (13% (*v/v*) ethanol) aged for 2–3 years, with acidities of 6.0 and 3.99%, respectively. These vinegars were fermented under controlled conditions at 20 °C in glass containers. Apple vinegar from Yecheon, Gyeongsangbuk-do, was manufactured using stationary fermentation at 30 °C in a traditional jar (pottery) and had an acidity of 6.56%. Korean blackberry (*Rubus coreanus*) and apple vinegar from Gochang, Jeollabuk-do, were fermented statically at 30 °C in traditional jars (pottery) and had lower acidities than other vinegars (5.8, 2.8, and 4.4%). The age of the collected vinegars ranged from 1 month to 1 year; they were produced in a farmhouse and then stored in an aging room after fermentation was complete (Table 1).

### 3.2. Identification of Isolates using 16S rRNA Sequencing

Acetic acid bacteria were successfully isolated from seven samples of fruit vinegar (other than tomato) using a CaCO$_3$ medium. The tomato vinegar sample did not yield any identifiable AAB colonies on the YGC screening medium. The tomato vinegar sample fermented at 20 °C for about 3 months showed an acidity of 4%; however, because AAB

could not be screened on the glucose-ethanol medium, it was recommended for their proper identification. Finally, a total of 256 presumptive AAB strains were isolated. Almost all isolates were capable of producing acid, indicated as a clear zone formed around all colonies on $CaCO_3$-ethanol agar. Alcoholic stress conditions were established on agar media with different ethanol concentrations. From the 66 isolates tested for acetic acid production in ethanol medium containing 3–15% ethanol using spot plate assay, 20 strains produced substantially higher acetic acid amounts than the control strains (*G. saccharivorans* CV1, *A. pomorum* 11998, and *A. syzygii* 12233) after 4 days of incubation in 3–15% ethanol medium. Almost all AAB tolerated ethanol concentrations up to 12% (Table 2 and Supplementary Figure S1). Seven strains showing excellent acid production over time in liquid cultures containing different ethanol concentrations along with 1% acetic acid were selected. The productivity of AAB strains isolated in this work was the highest, with 9% acetic acid production (Supplementary Table S1).

**Table 2.** Acetic acid production capacity observed as a clear zone formed around colonies on $CaCO_3$-ethanol agar.

| Alcohol Concentration | 3% | 5% | 7% | 9% | 10% | 12% | 15% |
|---|---|---|---|---|---|---|---|
| Strain | | | | | | | |
| KSO 5 [2] | 3.83 [1] | 3.98 | 3.72 | 3.21 | 3.33 | 3.70 | 2.60 |
| KSO 6 | 3.88 | 3.93 | 3.94 | 3.29 | 3.48 | 3.62 | 2.23 |
| KSF 2 | 3.41 | 3.17 | 1.58 | 0.00 | 0.00 | 0.00 | 0.00 |
| KSF 6 | 2.78 | 3.14 | 1.55 | 0.00 | 0.00 | 0.00 | 0.00 |
| KSF 8 | 2.70 | 3.32 | 2.16 | 0.00 | 0.00 | 0.00 | 0.00 |
| JGB 20-11 | 3.69 | 4.00 | 5.11 | 2.70 | 3.56 | 2.58 | 0.00 |
| JGB 20-13 | 3.56 | 4.18 | 3.41 | 2.48 | 3.98 | 2.02 | 0.00 |
| JGB 21-17 | 3.13 | 3.56 | 3.15 | 1.75 | 2.60 | 2.52 | 0.00 |
| JGB 21-20 | 2.79 | 3.23 | 2.97 | 2.37 | 3.89 | 3.16 | 1.58 |
| JGB 21-24 | 3.17 | 3.59 | 3.12 | 1.97 | 1.70 | 1.32 | 0.00 |
| JGA 10 | 3.04 | 3.46 | 0.00 | 0.00 | 0.00 | 0.00 | 0.00 |
| JGA 13 | 2.67 | 2.81 | 0.00 | 0.00 | 0.00 | 0.00 | 0.00 |
| JGA 16 | 2.74 | 2.32 | 0.00 | 0.00 | 0.00 | 0.00 | 0.00 |
| GHA 2 | 3.76 | 3.60 | 3.10 | 2.24 | 3.33 | 3.22 | 2.57 |
| GHA 7 | 3.65 | 3.58 | 3.22 | 2.24 | 4.24 | 3.08 | 3.53 |
| GHA 20 | 3.31 | 3.80 | 3.14 | 2.34 | 3.43 | 2.81 | 3.31 |
| GHA 112 | 2.94 | 3.55 | 3.26 | 2.07 | 3.30 | 2.70 | 2.38 |
| GYA 14 | 3.29 | 3.50 | 2.88 | 1.68 | 2.68 | 2.20 | 0.00 |
| GYA 17 | 3.39 | 3.30 | 2.84 | 1.92 | 3.04 | 2.28 | 2.24 |
| GYA 23 | 3.41 | 3.80 | 3.39 | 1.53 | 3.11 | 3.38 | 2.51 |
| GS CV1 [3] | 2.84 | 3.68 | 2.51 | 2.06 | 2.24 | 2.41 | 1.84 |
| AP 11998 | 1.20 | 1.10 | 1.09 | 0.00 | 0.00 | 0.00 | 0.00 |
| AS 12233 | 4.18 | 4.74 | 3.06 | 2.56 | 2.38 | 0.00 | 0.00 |

[1] Clear zone size (mm) using the potency index (PI) as a parameter. [2] The first capital letter is the abbreviation for the origin of the vinegar sample (see Table 1), and the second number is the number of the screened strain. [3] Symbols: GS CV1, *Gluconacetobacter saccharivorans* CV1, KACC 17057; AP 11998, *Acetobacter pomorum*, KACC 11998; and AS 12233, *A. syzygii*, KACC 12233.

Diversity was observed in the isolated AAB strains. Seven AAB strains were identified using 16S rRNA gene sequencing, with sequencing analysis revealing two different AAB species, *A. pasteurianus* and *A. cerevisiae* (Table 3).

The selected strains had more than 99% similarity based on 16S rRNA gene sequence analysis. The sequences were used to construct a phylogenetic tree that clearly showed the groupings of the samples and their relatedness within the *Acetobacteraceae* family (Figure 1). The two species identified here are commonly associated with vinegar production [33].

**Table 3.** Identification of acetic acid bacteria (AAB) using 16S rRNA with the basic local alignment search tool available in the National Center for Biotechnology Information database.

| Strain [1] | Size (bp) | Closest Match | Identity (%) |
|---|---|---|---|
| KSO 5 | 1411 | *A. cerevisiae* | 99.85 |
| JGB 20-11 | 1408 | *A. pasteurianus* | 99.78 |
| JGB 21-17 | 1409 | *A. pasteurianus* | 99.70 |
| JGB 21-20 | 1406 | *A. pasteurianus* | 99.85 |
| GHA 7 | 1409 | *A. pasteurianus* | 99.85 |
| GHA 20 | 1410 | *A. pasteurianus* | 99.85 |
| GYA 23 | 1409 | *A. pasteurianus* | 99.85 |

[1] The first capital letter is the abbreviation for the origin of the vinegar sample (see Table 1), and the second number is the number of the screened strain.

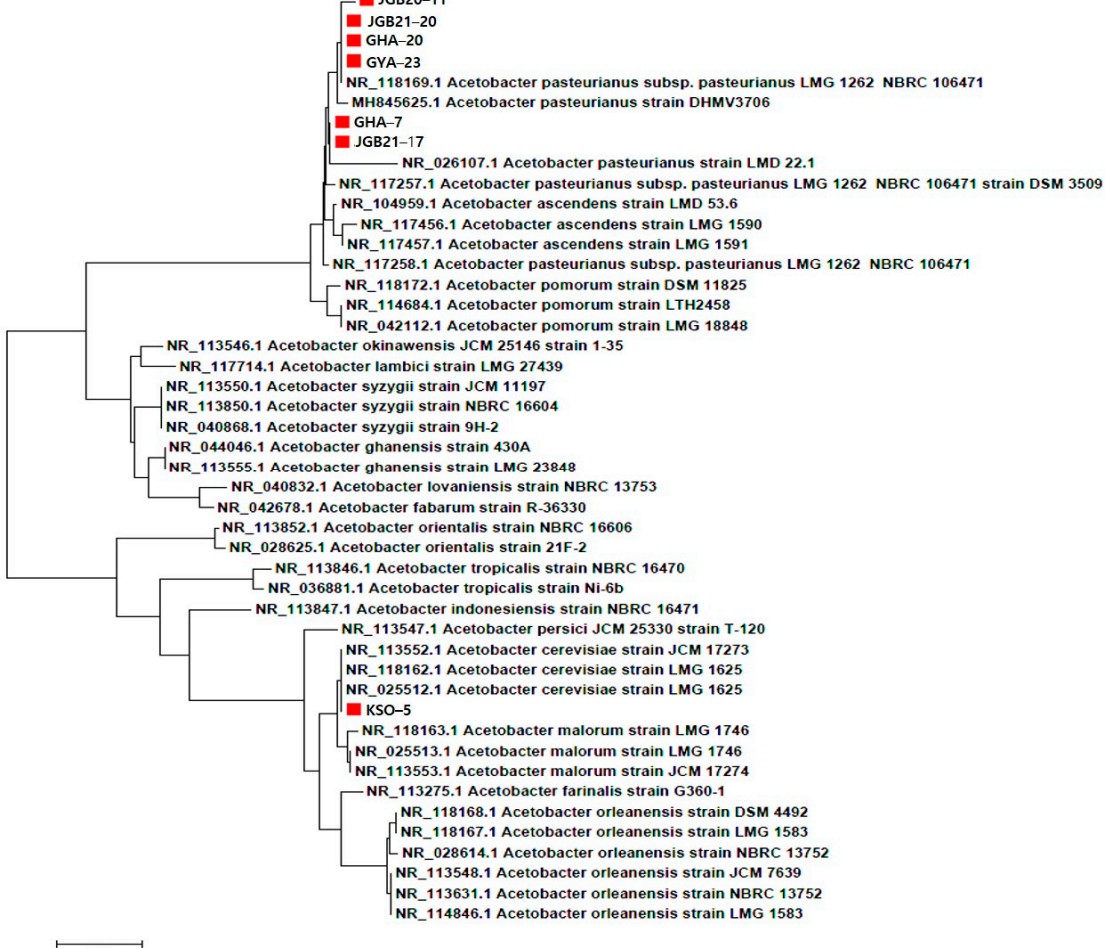

**Figure 1.** Phylogenetic relationships between the seven selected strains and other AAB species based on 16S rRNA gene sequences. The seven AAB strains selected in this work are marked with red rectangles.

### 3.3. Effect of Temperature on Growth

Temperature optimization is important for bacterial culture, as bacterial inactivation can occur above a fixed temperature due to the denaturation of essential enzymes, membrane damage with loss of cellular components, and increased sensitivity to the toxic effects of metabolic products, including acetic acid [34]. To select AAB strains suitable as starter strains for fermentation, we evaluated the growth rate and acetic acid yield of seven isolates at different temperatures (10–40 °C). All strains showed acidification capacity, with halos around agar colonies, but at different levels depending on the incubation temperature

across seven days (Figure 2). The seven AAB strains selected did not grow well at low temperatures (10 or 20 °C) due to a long adaptation time. As reported by Adachi et al. [35], AAB strains are usually mesophilic with optimum growth temperatures between 25 and 30 °C. The KSO 5, JGB 20-11, and JGB 21-20 strains showed a high acidification capacity at 30 °C, while the GHA 20 strain showed a capacity to grow at 40 °C. In terms of the acidification capacity at 40 °C, JGB 21-17, GHA 7, and GYA 23 strains had an acid zone diameter at 40 °C that was at least 90% of that observed at 30 °C. Acidification results from the bioconversion of ethanol to acetic acid, an exothermic reaction that can result in heat accumulation [36]. Increasing temperatures in recent years have posed a serious challenge to the fermentation industry because large cooling systems are required to maintain optimum temperatures. The production of vinegar using thermotolerant AAB has attracted interest because of its potential economic benefits. Therefore, AAB strains isolated in this study, which can grow at 40 °C, are expected to be highly useful in the vinegar industry as starter strains capable of strong acidification and improved fermentation.

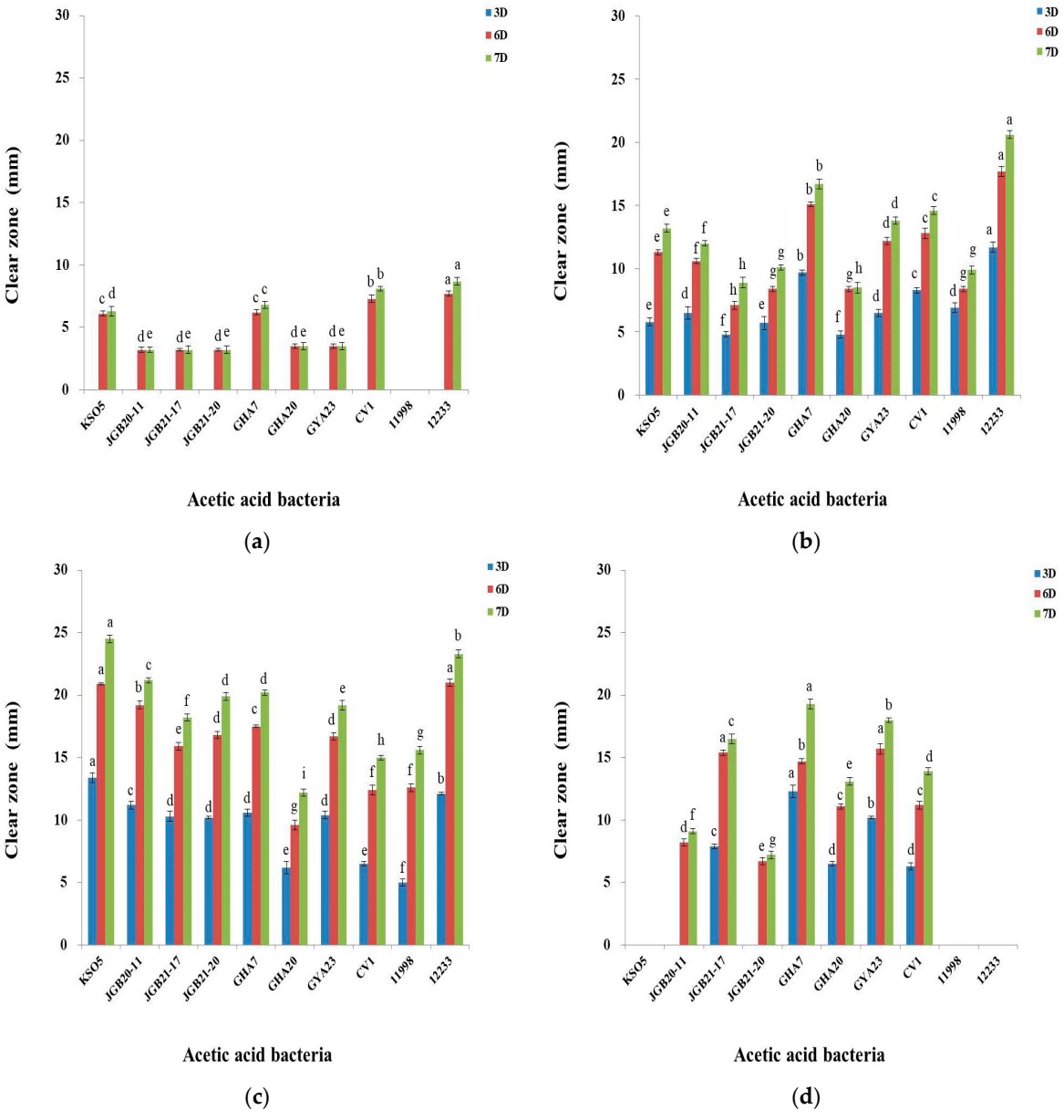

**Figure 2.** Growth characteristics of AAB at different temperatures; 10 °C (**a**), 20 °C (**b**), 30 °C (**c**), and

40 °C (**d**). The different letters above the error bars in the data for the same day are significantly different at $p < 0.05$, as determined by Duncan's multiple range test. 3D, 6D, and 7D are incubation periods of 3, 6, and 7 days, respectively. The three strains, CV1, *Gluconacetobacter saccharivorans* CV1; 11998, *Acetobacter pomorum*, KACC 11998; and 12233, *A. syzygii*, KACC 12233, were used as control strains.

### 3.4. Antibacterial Activity

Antibacterial effects of the AAB strains isolated against bacteria causing food spoilage were tested using diffusion assays. The results are presented in Table 4. All AAB strains showed antimicrobial activity against Gram-positive (*Bacillus cereus* and *Staphylococcus aureus*) and Gram-negative bacteria (*Escherichia coli* and *Salmonella typhimurium*). Comparing the antimicrobial effect of the strains with those for different concentrations of acetic acid, isolates showed an activity range comparable with 12.5–25 mg/mL acetic acid (Supplementary Figure S2 and Table S2). Antimicrobial activity was higher against Gram-positive bacteria, including *S. aureus*, than against Gram-negative bacteria. It has been reported that the outer membrane of Gram-positive bacteria is more sensitive to antimicrobial agents than that of Gram-negative bacteria [37]. Acetic acid produced by AAB can diffuse across bacterial membranes based on an equilibrium between ionized and non-ionized forms in parallel with the lowering of pH of the surrounding medium. Acidification of the cytoplasm causes morphological disruption of harmful bacteria and leakage of intracellular components, as well as the induction of protein unfolding along with membrane and DNA damage [38–40]. The antibacterial effect of acetic acid on different types of pathogenic bacteria has been increasingly relied upon in food safety for its preservative effects. Acetic acid is a substance generally recognized as safe by the Food and Drug Administration (FDA) and has been approved as a food additive by the European Commission, the Food and Agriculture Organization, the World Health Organization, and the FDA [41]. Sakhare et al. [42] investigated the use of acetic acid as an antimicrobial agent for meat, including poultry, beef, and pork, to extend its shelf life, as well as for the decontamination of bacteria, including *Salmonella* spp. and *E. coli*. We, therefore, assessed the possible use of antimicrobials derived from the AAB strains isolated here for application in the vinegar industry.

**Table 4.** Antibacterial effects of AAB against harmful organisms.

| Sample Type | Name | Titratable Acidity (%) | Clear Zone (mm) | | | |
|---|---|---|---|---|---|---|
| | | | Gram-Positive | | Gram-Negative | |
| | | | *B. cereus* | *S. aureus* | *E. coli* | *S. typhimurium* |
| Strain | KSO 5 | 4.17 | 17.6 ± 0.1 [g] [2] | 22.0 ± 0.1 [f] | 19.8 ± 0.1 [g] | 20.2 ± 0.2 [f] |
| | JGB 20-11 | 5.66 | 20.5 ± 0.1 [e] | 23.7 ± 0.2 [e] | 23.2 ± 0.1 [f] | 21.6 ± 0.1 [d] |
| | JGB 21-17 | 5.53 | 20.2 ± 0.2 [d] | 25.2 ± 0.1 [c] | 20.6 ± 0.1 [e] | 20.3 ± 0.2 [f] |
| | JGB 21-20 | 5.59 | 20.5 ± 0.1 [f] | 25.2 ± 0.2 [c] | 21.5 ± 0.1 [f] | 20.3 ± 0.2 [f] |
| | GHA 7 | 5.47 | 18.6 ± 0.2 [c] | 22.9 ± 0.1 [d] | 21.5 ± 0.2 [c] | 22.6 ± 0.1 [f] |
| | GHA 20 | 5.59 | 17.9 ± 0.2 [c] | 22.7 ± 0.1 [d] | 22.6 ± 0.2 [c] | 22.4 ± 0.1 [f] |
| | GYA 23 | 5.52 | 18.5 ± 0.1 [b] | 23.8 ± 0.2 [e] | 20.9 ± 0.1 [d] | 21.7 ± 0.1 [b] |
| Positive control [3] | AA 50 | | 27.3 ± 0.1 [a] | 31.1 ± 0.1 [a] | 30.1 ± 0.1 [a] | 30.1 ± 0.1 [a] |
| | AA 25 | | 20.2 ± 0.2 [f] | 24.1 ± 0.1 [b] | 24.2 ± 0.2 [b] | 22.1 ± 0.1 [c] |
| | AA 12.5 | | 16.2 ± 0.1 [h] | 16.1 ± 0.2 [d] | 17.1 ± 0.1 [f] | 17.2 ± 0.2 [e] |
| N.C. [1] | medium | | 8.1 ± 0.1 [i] | 8.1 ± 0.1 [g] | 8.1 ± 0.1 [h] | 8.1 ± 0.1 [g] |

[1] Negative control. [2] Means with different letters in the same column are significantly different, as indicated by Duncan's multiple range test ($p < 0.05$). [3] AA 12.5, AA 25, and AA 50 indicate acetic acid concentrations of 12.5, 25, and 50 mg/mL, respectively.

### 3.5. Antioxidant Activity

Antioxidants are known for their ability to limit radical reactions by transferring hydrogen atoms or electrons and by interrupting oxidative chain reactions. Assays using DPPH and ABTS measure the release of an electron to $ROO^\bullet$, converting it to an anion ($ROO^-$), resulting in the decrease in absorbance of the solution that reflects the concentration of the antioxidant. The mechanism involves the loss of a proton from the antioxidant, followed by electron transfer to the radical, which then reacts with the proton. This is influenced by proton affinity and electron transfer enthalpy. The DPPH radical reacts preferentially via a proton transfer mechanism in solvents such as ethanol and methanol, while the ABTS radical does the same in aqueous solutions.

The antioxidant properties of isolated AAB strains rely on the antioxidant molecules of AAB that have scavenging activity against free radicals [43]. The generation of free radicals leads to multiple chain reactions that can cause cell damage and death. The balance between free radicals and antioxidant molecules determines the level of oxidative stress [44]. Figure 3 shows the antioxidant activities of the isolated AAB strains measured using DPPH and ABTS. The scavenging capacity for DPPH and ABTS of the AAB strains was enhanced compared with that of the negative control (1% acetic acid). For the AAB strains, DPPH scavenging was approximately 50%, whereas that for the control was 22%. The value for the AAB strains was twice as high as that of the control and similar to that of 0.05% ascorbic acid (68.6%) (Figure 3a). The ABTS scavenging activity of the AAB strains was four times higher than that of the negative control (13.7%) and higher than that of ascorbic acid (49.7%; Figure 3b). Although scavenging of DPPH and ABTS radicals was different due to a difference in reactivity between the two compounds, the isolated AAB strains showed a correlation between antioxidant activity and organic acid levels, as previously reported by Xu et al. [43]. These findings suggest that acetic acid fermentation with various food materials increases their bioactive potential and promotes synergy between fermentation metabolites and microorganisms, leading to the production of compounds of interest, such as polyphenols. These results represent preliminary findings that will aid in the evaluation of the bioactive potential of acetic acid fermentation by isolated AAB strains.

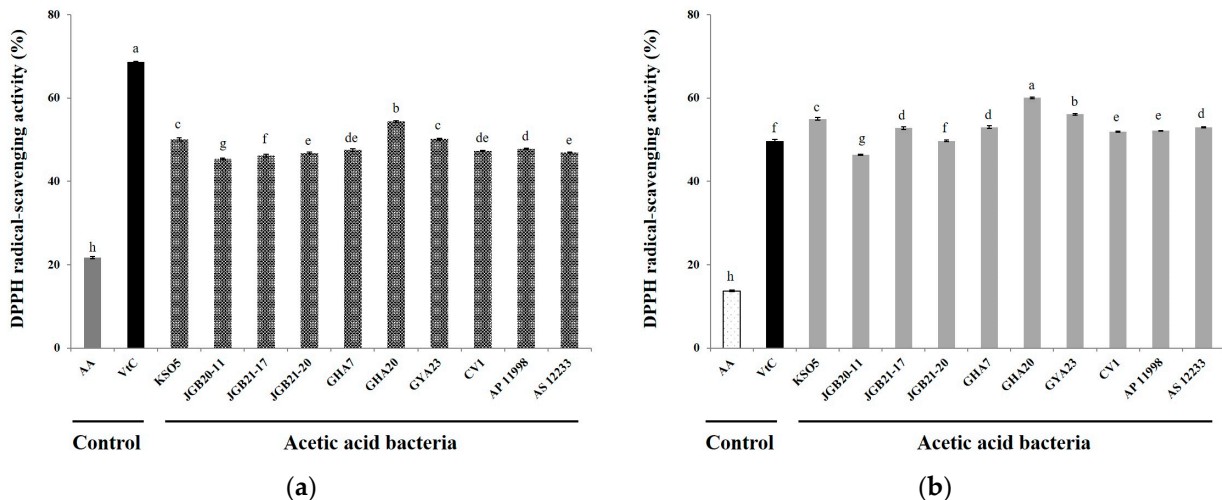

**Figure 3.** Antioxidant activities of isolated AAB strains determined as DPPH (**a**) and ABTS (**b**) free radical scavenging activity (%). Values are shown as mean ± SD (*n* = 3). Different letters (a–h) represent significant differences indicated by Duncan's multiple range test (*p* < 0.05). AA, acetic acid; VtC, vitamin C (ascorbic acid).

### 3.6. ACE Inhibition

Angiotensin-converting enzyme is a key enzyme in the renin-angiotensin-aldosterone system, converting angiotensin I to angiotensin II (Ang II) to exercise blood pressure control. Ang II binds to A-II receptors, constricts arteries and arterioles, excites the adrenal cortex,

and promotes aldosterone release. Ultimately, this causes an increase in blood pressure [45]. Inhibition of ACE alleviates high blood pressure by minimizing Ang II formation. Captopril is an effective synthetic antihypertensive drug that inhibits ACE [46]. Figure 4 shows ACE inhibition by culture supernatants of the isolated AAB strains. Six of the isolated strains (excluding GYA 23) had higher ACE inhibitory activity than the 0.1% captopril-positive control (76.9%). The KSO 5 strain showed the highest level of inhibition (91.3%). For acetic acid, the main product of AAB, ACE inhibitory activity was 58 and 96.0% at 12.5 and 25 mg/mL acetic acid concentrations, respectively. Some of the beneficial effects of acetic acid produced by fermentation have been attributed to ACE inhibition. Acetic acid is a carboxylic acid, and some carboxylic acids, including citric, docosahexaenoic, and tartaric acids, are known for their antihypertensive activity [47–50]. The ACE2 receptor is a high-affinity receptor for the viral spike protein of the severe acute respiratory syndrome coronavirus 2 (SARS-CoV-2) [51]. For this reason, ACE inhibitors and Ang II receptor blockers have been considered for the treatment of viral infections. As synthetic ACE inhibitors are effective antihypertensive drugs, they may cause adverse effects. Thus, there is a growing interest in identifying ACE inhibitors in natural products as alternatives to synthetic drugs. Vinegar may play a role in lowering blood pressure, and various studies have tested this effect [52,53]. Based on the results shown in Figure 4, these strains are expected to be highly useful sources of ACE inhibitors.

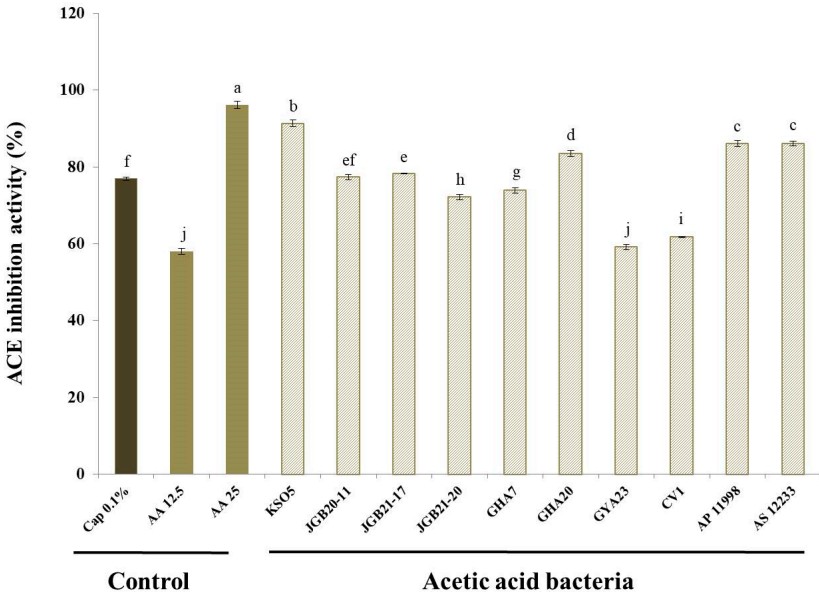

**Figure 4.** Angiotensin-converting enzyme inhibition by AAB supernatants. Values are shown as mean ± SD (*n* = 3). The different letters represent significant differences indicated by Duncan's multiple range test (*p* < 0.05). Cap 0.1%, 0.1% (*w/v*) captopril; AA 25, acetic acid 25 mg/mL; AA 12.5, acetic acid 12.5 mg/mL.

*3.7. α-Glucosidase Inhibition*

α-D-Glucosidase is a glucohydrolase that acts on (1→4) glucosidic bonds and is located in the brush border of the small intestine. Hydrolysis of terminal, non-reducing (1→4)-linked α-D-glucose units results in the release of D-glucose. The liberated glucose is absorbed in the gut, resulting in postprandial hyperglycemia. Inhibition of intestinal α-glucosidase can significantly decrease the postprandial increase in blood glucose levels after carbohydrate intake by delaying carbohydrate hydrolysis and absorption. Therefore, inhibition of this enzyme can be important in the management of hyperglycemia linked to type 2 diabetes [54–56]. Some antidiabetic drugs inhibit α-glucosidase activity. Although efficient in suppressing the rise in blood glucose levels in many patients, there are undesirable side effects associated with the continuous use of antidiabetic drugs [57,58].

Thus, natural inhibitors of α-glucosidase with no adverse or unwanted secondary effects are required.

The effects of AAB on α-glucosidase inhibition are presented in Figure 5. The inhibition capacity of AAB was determined to be 103% of that for 12.5 mg/mL acetic acid (AA 12.5) used as the positive control. Most strains showed inhibitory activity greater than that of the positive control, with the exception of 98.3% inhibition for the JGB 21-17 and GHA 7 strains. Acetic acid bacteria are candidate strains with good functionality and potential health benefits. For example, the α-glucosidase inhibitory ability of Kujippong (*Cudrania tricuspidata*) vinegar produced using AAB was previously shown to be 91.4% (after 72 h fermentation) [59]. Acetic acid, the main component produced by AAB, breaks down lactic acid to relieve fatigue and decomposes fat to help control weight. Weight control contributes significantly to the improvement of blood glucose levels [60].

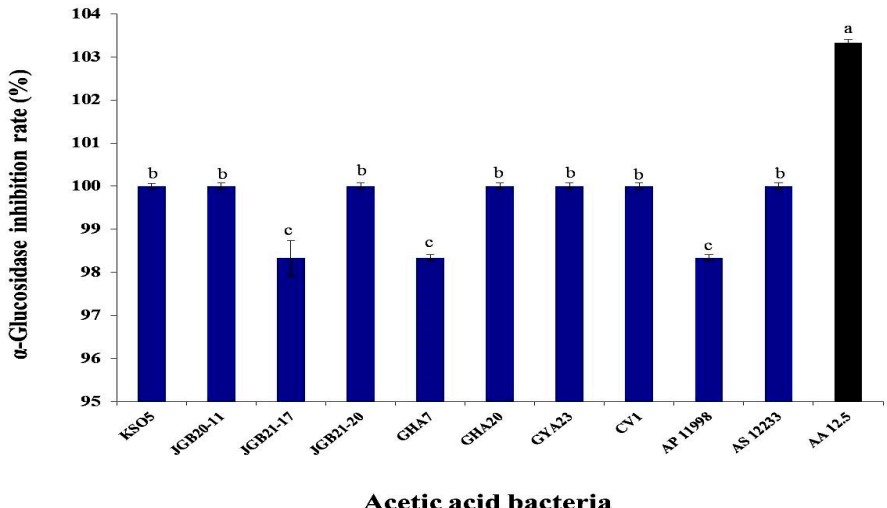

**Figure 5.** α-Glucosidase inhibition by isolated AAB strains. Values are shown as mean ± SD (*n* = 3). Different letters indicate significant differences using Duncan's multiple range test (*p* < 0.05). AA 12.5 indicates 12.5 mg/mL acetic acid.

## 4. Conclusions

In this study, we evaluated the functional characteristics of seven AAB strains isolated from farm-produced fermented fruit vinegars. Previously, we isolated *A. pasteurianus* strains from grain vinegar [28]. However, the strains isolated in this study from fruit vinegar were more diverse and included *A. cerevisiae* and *A. pasteurianus*. The strains isolated in this study produced 9% more acetic acid and exhibited 12% higher resistance to alcohol. Acidification of GHA 7, GYA 23, JGB 21-17, and GHA 20 strains at a growth temperature of 40 °C is expected to be useful for the vinegar industry. The bioactivities of these isolated strains, including their antibacterial, antioxidant, antihypertensive, and antidiabetic effects, suggest usefulness in the industrial sector as well as in the production of functional foods. By using excellent indigenous GHA 20 strain in the vinegar industry, we want to increase the utilization value of domestic AAB and reduce production costs. We have successfully demonstrated bioactive characteristics that may facilitate the use of these AAB as seed strains for the high-efficiency production of functional vinegar by rationally harnessing their functional characteristics.

**Supplementary Materials:** The following supporting information can be downloaded from https://www.mdpi.com/article/10.3390/fermentation9050447/s1. Figure S1: Ability of AAB isolates to produce acetic acid in CaCO$_3$ medium with different levels of ethanol (3, 5, 7, 9, 10, 12, and 15% (*v*/*v*)). Figure S2: Images of clear zones and calibration curves of acetic acid indicating quantitative antibacterial activity of selected AAB strains. Table S1: Acetic acid production over a time course of

days in a liquid medium containing 5% (*v/v*) ethanol and 1% (*v/v*) acetic acid. Table S2: Quantitative antibacterial activity of selected AAB strains using the calibration curve of acetic acid.

**Author Contributions:** S.-H.K. and S.-H.Y.: designed the study. S.-H.K.: performed the statistical analysis, wrote the protocol, and wrote the first draft of the manuscript. W.-S.J. and S.-Y.K.: managed the analyses of the study and literature searches. All authors have read and agreed to the published version of the manuscript.

**Funding:** This research was funded by the Research Program for Agricultural Science and Technology Development (Project No. PJ01416103) and the National Institute of Agricultural Science, Rural Development Administration, Republic of Korea.

**Institutional Review Board Statement:** Not applicable.

**Informed Consent Statement:** Not applicable.

**Data Availability Statement:** Data are contained within the article.

**Acknowledgments:** The financial support of the Rural Development Administration is gratefully acknowledged.

**Conflicts of Interest:** The authors declare no conflict of interest.

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
