# Peer review of "Quality and Functional Characterization of Acetic Acid Bacteria Isolated from Farm-Produced Fruit Vinegars"

_fermentation, doi:10.3390/fermentation9050447_

Round 1

Reviewer 1 Report

The design of this paper is relatively reasonable. The authors evaluated the functional characteristics of seven AAB strains isolated from farm-produced fermented fruit vinegars. But there are still some details that need improvement

Abstract: The abstract section was not concise enough and was a bit verbose

Line11: ‘Acetobacter cerevisiae and A. pasteurianus?’ Please unify the abbreviations for the entire text

Line 12: ‘Physiological properties were determined, including ethanol tolerance at up to 12% (v/v).‘ Inaccurate expression.

Introduction: The logic is not clear enough, and there is a lack of connecting sentences between paragraphs.

Table 1: Please remove the dashed line

Line 165: Should there be spaces before and after the multiplication sign?

Line 168: In the formula, it should be ‘??? ??â„Ž??????? rate’. Should there be a space before the minus sign?

Line 179-180:α - Glucosidase inhibition’ or ‘α-Glucosidase inhibition?’

Line 179-180: In the formula, it should be ‘inhibition rate’

Figure 2: ‘Values are expressed as mean ± standard deviation (3 replicates) range test (p<0.05).’ Inaccurate expression. How did the author conduct a significance analysis

Line 348: 3.7.α-. Glucosidase Inhibition. Is there an extra point?

Conclusions: Should be more concise, please rewrite

Author Response

Responses to the comments from reviewer #1

â—‡ Journal: Fermentation

â—‡ Manuscript ID: fermentation-2337562

â—‡ Title: Quality and Functional Characterization of Acetic Acid Bacteria Isolated from Farm-Produced Fruit Vinegars

Dear Editor,

We would like to express our gratitude to the reviewer for the valuable comments on our manuscript. Our responses to the reviewer’s comments are given below. The relevant changes in the manuscript are marked using the “Track Changes” option in Word processor.

  1. Does the introduction provide sufficient background and include all relevant references? : Can be improved

→ We have tried to improve the Introduction section as suggested by the reviewer. Please see our responses to specific comments below.

  1. Are all the cited references relevant to the research? : Yes

→ We thank the reviewer for the positive comment.

  1. Is the research design appropriate? : Yes

→ We thank the reviewer for the positive comment.

  1. Are the methods adequately described? : Yes

→ We thank the reviewer for the positive comment.

  1. Are the results clearly presented? : Can be improved

→ We have improved the Results section as suggested by the reviewer. Please see our responses to specific comments below.

  1. Are the conclusions supported by the results? : Must be improved

→ We have improved the Conclusions section as suggested by the reviewer. Please see our responses to specific comments below.

Comments and Suggestions for Authors

  1. Abstract: The abstract section was not concise enough and was a bit verbose.

→ We have edited the Abstract for brevity. We have revised the sentence in lines 12–16.

  1. Line 11: ‘Acetobacter cerevisiae and A. pasteurianus?’ Please unify the abbreviations for the entire text.

→ Because these names are mentioned for the first time in this line, we have given the full name.

  1. line 12: ‘Physiological properties were determined, including ethanol tolerance at up to 12% (v/v).’ Inaccurate expression.

→ We apologize for the inaccurate expression. We have modified the sentence to read as follows: “These strains were tolerant to ethanol concentrations up to 12% (v/v).”

  1. Introduction: The logic is not clear enough, and there is a lack of connecting sentence between paragraphs.

→ We are sorry for the lack of clarity. For a better description of the logic, we have included information on the current state of knowledge on the isolation of acetic acid bacteria for use in acetic or alcoholic fermentation of food products (lines 49–51). We have also cited additional references.

  1. Table 1: Please remove the dashed line.

→ We have removed the dashed line, as suggested.

  1. Line 165: Should there be spaces before and after the multiplication sign?

→ We have deleted the spaces before and after the multiplication sign.

  1. Line 168: In the formula, it should be ‘ACE inhibition rate’. Should there be a space before the minus sign?

→ We have deleted the space before the minus sign and have changed the term to “ACE inhibition rate” in the formula.

  1. Line 179-180: ‘α - Glucosidase inhibition’ or ‘α-Glucosidase inhibition’?

→ We have corrected the term to “α-Glucosidase inhibition” at all places in the manuscript.

  1. Line 179-180: In the formula, it should be ‘inhibition rate’.

→ We corrected the term to “α-Glucosidase inhibition rate” in the formula.

  1. Figure 2: ‘Values are expressed as mean ± standard deviation (3 replicates) range test (p<0.05).’ Inaccurate expression. How did the author conduct a significance analysis.

→ We have revised the sentence to read as follows: “The different letters above the error bars in the data for the same day are significantly different at p < 0.05, as determined by Duncan’s multiple range test.

  1. Line 348: 3.7.α-. Glucosidase inhibition. Is there an extra point?

→ There was no an extra period in the manuscript file that we uploaded. Nonetheless, we have checked it again and ensured that the heading reads as follows: 3.7. α-Glucosidase Inhibition.

  1. Conclusions: Should be more concise, please rewrite.

→ We have rewritten the conclusions.

We are thankful to the reviewer for the detailed comments that have helped us improve the manuscript. We have addressed all the concerns. We believe that the changes made in the revised manuscript have strengthened it and hope that the revised manuscript is suitable for publication in Fermentation.

Sincerely,

Soo-Hwan Yeo

Reviewer 2 Report

The article's introduction needs to be corrected. There is a lack of transfer of current knowledge on the isolation of microorganisms responsible for acetic or alcoholic fermentation from food products. The last paragraph needs to be rewritten because it should include the rationale for undertaking the research along with the purpose and scope of the work. The authors in this part of the article should not summarize it and give conclusions - this should be done in the final chapter.

Table 1 - The authors should write instead of “Rubus coreanus (Bokbunja3)_20” what little say to the reader, the name “Korean blackberry (Rubus coreanus)”. The same situation is for “Schisandra chinesis (Omija4)” – instead should be “Magnolia berry (Schisandra chinensis)”.

Chapter 3.1 should be merged with Chapter 2.1 and the chapter should be renamed after the merger.

line 204-205- The authors should expand on this sentence and explain why it was not possible to isolate bacterial strains from tomato vinegar.

Table2 and 3- The authors should state what the names of the tested isolates consist of. They should refer to the letters and numbers in the names of the strains.

line 205- The authors say they isolated 256 strains of AAB bacteria, and according to Tables 2 and 3, they tested only some of them. They need to write in the text why they chose only some of them for testing and why.

Figure 2- the question of why only 7 strains were selected for testing and identified becomes legitimate again. In addition, you should indicate which strains are control and which are test strains in the footer of the figure.

Conclusions- The conclusions are concise and substantive and are based on the authors' research. I would only add which of the 7 strains identified was the best and the authors recommend it for industrial use.

Author Response

Responses to the comments from reviewer #2

â—‡ Journal: Fermentation

â—‡ Manuscript ID: fermentation-2337562

â—‡ Title: Quality and Functional Characterization of Acetic Acid Bacteria Isolated from Farm-Produced Fruit Vinegars

Dear Editor,

We would like to express our gratitude to the reviewer for the valuable comments on our manuscript. Our responses to the reviewer’s comments are given below. The relevant changes made in the manuscript are marked using the “Track Changes” option in Word processor.

  1. Does the introduction provide sufficient background and include all relevant references? : Must be improved

→ We have tried to improve the Introduction section as suggested by the reviewer. Please see our responses to specific comments below.

  1. Are all the cited references relevant to the research? : Yes

→ We thank the reviewer for the positive comment.

  1. Is the research design appropriate? : Yes

→ We thank the reviewer for the positive comment.

  1. Are the methods adequately described? : Can be improved

→ We have tried to improve the Methods section as suggested by the reviewer. Please see our responses to specific comments below.

  1. Are the results clearly presented? : Can be improved

→ We have improved the Results section as suggested by the reviewer. Please see our responses to specific comments below.

  1. Are the conclusions supported by the results? : Can be improved

→ We have improved the Conclusions section as suggested by the reviewer. Please see our responses to specific comments below.

Comments and Suggestions for Authors

  1. The article’s introduction needs to be corrected. (1) There is a lack of transfer of current knowledge on the isolation of microorganisms responsible for acetic or alcoholic fermentation from food products. (2) The last paragraph needs to be rewritten because it should include the rationale for undertaking the research along with the purpose and scope of the work. The authors in this part of the article should not summarize it and give conclusions-this should be done in the final chapter.

→ (1) As suggested, we have included information on the current state of knowledge on the isolation of acetic acid bacteria for use in acetic or alcoholic fermentation of food products in the Introduction section (lines 52–54). We have also cited additional references. (2) We have deleted the last paragraph from the Introduction section and relocated it to the Conclusions section.

  1. Table 1 – The authors should write instead of “Rubus coreanus (Bokbunja3)_20” what little say to the reader, the name “Korean blackberry (Rubus coreanus)”. The same situation is for “Schisandra chinesis (Omija4)”- instead should be “Magnolia berry (Schisandra chinensis)”.

→ As suggested, we have changed the sample names.

  1. Chapter 3.1 should be merged with Chapter 2.1 and the chapter should be renamed after the merger.

→ We have merged sections 2.1 and 3.1. The revised sections are renamed as follows: 2.1. Collection and Preparation of Samples; 3.1. Identification of Collected Vinegars.

  1. line 204-205 – The authors should expand on this sentence and explain why it was not possible to isolate bacterial strains from tomato vinegar.

→ We have explained why it was not possible to isolate bacterial strains from tomato vinegar (lines 226–230).

  1. Table 2 and 3 – The authors should state what the names of the tested isolates consist of. They should refer to the letters and numbers in the names of the strains.

→ We explained the letters and numbers in the names of the strains in the footnotes to Table 2 and 3.

  1. Line 205 – The authors say they isolated 256 strains of AAB bacteria, and according to Tables 2 and 3, they tested only some of them. They need to write in the text why they chose only some of them for testing and why.

→ We have included the screening method for the spot plate assay in the Material and Methods section for further clarity (lines 234–240).

  1. Figure 2 – the question of why only 7 strains were selected for testing and identified becomes legitimate again. In addition, you should indicate which strains are control and which are test strains in the footer of the figure.

→ We have explained why only seven strains were selected for testing in section 3.2 (lines 238–241).

  1. Conclusions – The conclusions are concise and substantive and are based on the authors’ research. I would only add which of the 7 strains identified was the best and the authors recommend it for industrial use.

→ As suggested, we have rewritten the Conclusions section including the information sought by the reviewer.

We are thankful to the reviewer for the detailed comments that have helped us improve the manuscript. We have addressed all the concerns. We believe that the changes made in the revised manuscript have strengthened it and hope that the revised manuscript is suitable for publication in Fermentation.

Sincerely,

Soo-Hwan Yeo

Reviewer 3 Report

This manuscript is about the characterization of some acetic acid bacteria isolated from farm-produced fruit vinegars. The topic is meaningful, but the manuscript in its current state is more like a work report than an academic paper. It can provide little useful information for readers.

Other comments:

1. More detailed information about the vinegar samples should be provided, like the fermentation stage form which the samples were taken.

2. Usually the microbial community information of different fermentation should be known.

3.The authors should discuss the differences between their results and those previously reported. If there are nothing new, the meaning of the study is limited.

Author Response

Responses to the comments from reviewer #3

â—‡ Journal: Fermentation

â—‡ Manuscript ID: fermentation-2337562

â—‡ Title: Quality and Functional Characterization of Acetic Acid Bacteria Isolated from Farm-Produced Fruit Vinegars

Dear Editor,

We would like to express our gratitude to the reviewer for the valuable comments on our manuscript. Our responses to the reviewer’s comments are given below. The relevant changes in the manuscript are marked using the “Track Changes” option in Word processor.

  1. Does the introduction provide sufficient background and include all relevant references? : Yes

→ We thank the reviewer for the positive comment.

  1. Are all the cited references relevant to the research? : Yes

→ We thank the reviewer for the positive comment.

  1. Is the research design appropriate? : Can be improved

→ We have improved the description of the research design. Please see our responses to specific comments below.

  1. Are the methods adequately described? : Can be improved

→ We have improved the description of the methods. Please see our responses to specific comments below.

  1. Are the results clearly presented? : Can be improved

→ We have improved the Results section as suggested by the reviewer. Please see our responses to specific comments below.

  1. Are the conclusions supported by the results? : Can be improved

→ We have improved the Conclusions section as suggested by the reviewer. Please see our responses to specific comments below.

Comments and Suggestions for Authors

  1. More detailed information about the vinegar samples should be provided, like the fermentation stage form which the samples were taken..

→ We have summarized the information about the period of sample production and collection in Table 1.

  1. Usually the microbial community information of different fermentation should be known..

→ A combination of culture-independent and culture-dependent methods is recommended as an effective method to overcome the difficulties in the isolation and cultivation of AAB strains. The fundamental limitation in ecological studies remains the difficulty in isolation, cultivation, and maintenance of pure cultures. In spite of the limitations, the methods that have been developed to characterize the genetic variation in AAB are based on culture-dependent techniques, as there is currently no alternative to bypass the culture step. Furthermore, we performed the culture-dependent methods because we had to develop a database of indigenous AAB and evaluate the possible use of AAB strain as a starter culture for vinegar production.

  1. The authors should discuss the differences between their results and those previously reported. If there are nothing new, the meaning of the study is limited.

→ Previously, we isolated A. pasteurianus strains from grain vinegar; the strains isolated from fruit vinegar were more diverse and included A. cerevisiae and A. pasteurianus. The strains isolated in this study produced 9% more acetic acid and exhibited 12% higher resistance to alcohol. Acidification of GHA 7, GYA 23, JGB 21-17, and GHA 20 strains at a growth temperature of 40 °C is expected to be useful for the vinegar industry. We have emphasized this fact in the Conclusions section.

We are thankful to the reviewer for the detailed comments that have helped us improve the manuscript. We have addressed all the concerns. We believe that the changes made in the manuscript have strengthened it and hope that the revised manuscript is suitable for publication in Fermentation.

Sincerely,

Soo-Hwan Yeo

Round 2

Reviewer 2 Report

The authors have addressed all comments and concerns. Based on the analysis of the changes made in the introduction, methodology and description of results sections, I conclude that the authors have significantly improved the quality of the article. Also the tables and graphs and the information presented on them have been improved and better explained. The conclusions have been significantly improved and supplemented with the necessary information.

Reviewer 3 Report

The revisions are almost acceptable.